# Evaluation of the *Longissimus Thoracis et Lumborum* Muscle Quality of Chaka and Tibetan Sheep and the Analysis of Possible Mechanisms Regulating Meat Quality

**DOI:** 10.3390/ani13152494

**Published:** 2023-08-02

**Authors:** Xianli Xu, Hongjin Liu, Xungang Wang, Qian Zhang, Tongqing Guo, Linyong Hu, Shixiao Xu

**Affiliations:** 1Northwest Institute of Plateau Biology, Chinese Academy of Sciences, Xining 810008, China; ucasxianlixu@163.com (X.X.); guotongqing2022@163.com (T.G.); 2University of Chinese Academy of Sciences, Beijing 100049, China

**Keywords:** Sheep, M. *longissimus thoracis et lumborum*, meat quality, transcriptomics, metabolomics

## Abstract

**Simple Summary:**

The quality of mutton is closely related to the development of the entire animal husbandry and sheep industry. The present work quantified the difference in meat quality between Chaka (CK) sheep and (Tibetan) TB sheep through the technology of transcriptomics, combined with metabolomics, to identify genes and metabolites, which are related to meat quality. Our study found that TB sheep have better nutritional value, while CK sheep have better flavor. The *GPAT2*, *PLPP2*, *AGPAT1*, *PNPLA2*, and *GPAT4* genes were identified as key genes that regulate glycerophospholipid metabolism and correlate with meat color, texture, and flavor. Our study comprehensively compared the *longissimus thoracis et lumborum* (LTL) muscle of CK sheep and TB sheep and explored the difference in meat quality at the molecular level, which may provide a strategy for improving meat quality from a molecular perspective.

**Abstract:**

This study aimed to comprehensively evaluate the characteristics in the *longissimus thoracis et lumborum* (LTL) muscle of Chaka (CK) sheep and Tibetan (TB) sheep, and transcriptomics–metabolomics association analysis was used to find the possible genes, differential metabolites, and significant differential metabolic pathways that lead to meat quality differences. Based on the researched results, the nutritional quality of meat, including the contents of ether extract (11.95% vs. 10.56%), unsaturated fatty acid (51.20% vs. 47.69%), and polyunsaturated fatty acid (5.71% vs. 3.97%), were better in TB sheep than in CK sheep, while the CK sheep has better muscle fiber characteristics, such as the total number (62 vs. 45) and muscle fiber density (1426.54 mm^2^ vs. 1158.77 mm^2^) and flavor. Omics research has shown that the key differential metabolites and metabolic pathways were dominated by amino acid metabolism, particularly the glutathione metabolism, taurine and hypotaurine metabolism, and lipid metabolism-related pathways, such as glycerophospholipid metabolism and the sphingolipid signaling pathway. The intermediate metabolite sn-Glycerol 3-phosphoethanolamine played a key role in determining sheep meat quality, which was regulated by *GPAT2*, *PLPP2*, *AGPAT1*, *PNPLA2*, and *GPAT4* and correlated with meat color, texture, and flavor. Overall, these results will provide effective information and more evidence to support further exploration of valuable biomarkers of meat quality.

## 1. Introduction

With the development of the social economy and the rapid improvement in people’s living standards, people attach great importance to a healthy and reasonable diet. The demand for food has changed from “quantity” to “quality”, livestock and poultry meat products are seen as an important source of animal protein, and their quality and nutritional value have received ever-more attention [1,2]. The quality of the meat is the crucial factor influencing a consumer’s choice, which depends on comprehensive sensory attributes, including meat color, flavor, taste, and other nutritional values [3].

The animal breed is a key factor that determines the meat quality because each species contains different genes that control meat traits and the physical and chemical composition of the meat quality. Previous studies have shown that animal breeds have an impact on meat flavor and composition [4]. By comparing the meat quality of the *longissimus dorsi* muscles of three different breeds of cattle (Yunling cattle, Wenshan cattle, and Simmental cattle), Meng et al. [5] found that there were differences in meat quality, such as meat color, pH, muscle tenderness, and cooking loss, in these three breeds. Through comparative analysis of the carcass and meat characteristics of Chinese Mongolian sheep and Dober sheep, Xiang et al. [6] found that the carcass traits of Dober sheep, such as lean meat weight, slaughter rate, and the ratio of lean meat to live weight, were significantly higher than those of Chinese Mongolian sheep, but its muscle nutritional value was lower. Chaka sheep originate from Wulan County, Haixi Mongolian and Tibetan Autonomous Prefecture, Qinghai Province. As a semi-fine wool sheep originating from the Qinghai–Tibet Plateau, Chaka sheep is listed as a valuable breed of the Qinghai–Tibet Plateau because of its tender meat, fat but not greasy, fresh but not fatty, and rich in mineral and vitamin characteristics, which received the National Geographical Indication in 2013 [7]. Tibetan sheep is one of the three original sheep breeds in China, and it is native to Tibet, China, though it can also be found in Qinghai, the northwest of Sichuan, and the south of Gansu. It has the characteristics of cold resistance, rough feeding resistance, high altitude adaptability, robust physique, and rapid action, and it not only produces fur, but also has good meat value, and it is a distinctive resource breed with great development value [8]. Chaka sheep and Tibetan sheep are famous sheep breeds on the Qinghai–Tibet Plateau and have perfect meat production characteristics. However, there are few studies about the comparison between the difference in meat quality between the two characteristic varieties and the mechanism that may regulate corresponding differential meat quality.

Nowadays, multi-omics linkage techniques are widely used in the muscle tissues of sheep, cattle, and pig to uncover the molecular mechanisms associated with muscle metabolism [9], which has the advantage of providing comprehensive biological information and promoting a deeper understanding of biological systems, but has the disadvantages of complexity in data processing and analysis, high cost and technology requirements, and limitations in sample quantity and quality [10]. Studies have shown that transcriptomics and metabolomics are powerful tools for studying meat quality traits [11]. For example, Ramanathan et al. [12] studied the meat color and pH changes in the longest back muscle of dark-cutting beef. Identification of key genes regulating important muscle flavor precursors in sheep using transcriptome and metabolome was performed by Zhang et al. [13]. Differentially expressed genes (DEGs) are obtained through transcriptomic studies, while metabolomic studies can identify the corresponding differential metabolites and differential metabolic pathways, and molecular network interactions of differentially expressed genes, metabolites, and metabolic pathways are established through correlation analysis of transcriptomics and metabolomics [14]. This process allows further comprehensive analysis of the complex molecular mechanisms regulating meat quality.

The present study assessed the meat quality and muscle transcriptome and metabolome of Chaka sheep and Tibetan sheep. There were three main purposes of this study. Firstly, we expected to extensively evaluate the value of these two breeds by comparing the meat quality. Secondly, we intended to identify the specific and superior meat quality indicators based on the horizontal and vertical comparisons and identify the specific superior phenotypes. Finally, through transcriptomics and metabolomics correlation analysis, we hoped to identify potential muscle bio-markers for mutton quality and provide a theoretical basis for sheep breed selection and the cultivation of new high-quality meat sheep breeds.

## 2. Materials and Methods

### 2.1. Experimental Animals

In this study, six female 1-year-old Chaka and Tibetan sheep weighing 55.56 ± 3.89 kg and 57.51 ± 4.62 kg, respectively, were selected under the same natural grassland (Chaka Town, Wulan County, Qinghai Province, China) grazing conditions. Approximately 350 g of LTL samples were collected from between the 12th and 13th ribs of each sheep. LTL tissue was immediately frozen in liquid nitrogen and transported to the laboratory for storage at −80 °C.

### 2.2. Color Measurement and pH Value

The specimens were exposed to air for 30 min at 4 °C (bloom), and the color was measured using an OPTO-LAB (Konica Minolta, Inc., Tokyo, Japan) meat color analyzer. The pH of each sample was measured using a PH-STAR Carcass Muscle PH Value Direct Tester (MATTHAUS GmBh, Bavaria, Portmeuse, Germany). Each sample from the LTL was measured in triplicate.

### 2.3. Analysis of Chemical Components and Muscle Fiber Parameters

AOAC standard procedures were used to evaluate the content of crude protein (CP), moisture (MSTR), total ash (Ash), and ether extract (EE). The MSTR content of the meat samples was determined by drying the meat samples in an oven at 105 °C. The CP and EE contents were estimated using the Soxhlet extraction method and the Kjeldahl technique, respectively. The meat samples were burned in a crucible at a temperature of 550 °C for 4 h to determine the content of ash.

The traditional hematoxylin and eosin (H&E) staining method was used to measure the total number, total area, density, and diameter of the muscle fibers. Slides were imaged via a Leica RM2235 slide scanner (Leica RM2235, Nussloch, Germany), and images were acquired using CaseViewer 6.0 software (3DHistech, Ltd., Budapest, Hungary).

### 2.4. Determination of Amino Acids and Fatty Acids

Amino acids were determined via a Sykam S-433D Amino Acid Analyzer. Next, 25 mg of dry powder of each meat sample was weighed and hydrolyzed for 22–24 h, we removed 1 mL of sample dilution to the test tube concentrator, and 5 mL of sample diluent was added and mixed well. The mixed sample was aspirated using a disposable syringe and filtered into the injection vial using a needle filter. The chromatographic conditions were as follows: column: Na+ sulfonic cation exchange resin; column temperature: 58 °C; reactor temperature: 130 °C; detection wavelengths: 570 nm and 440 nm.

GC-MS technology was used to detect the fatty acids, and 5 g of meat sample was mixed well using a vortex mixer, aspirated using a disposable syringe, and filtered into the injection vial using a needle filter. Then, 8 mL of sodium hydroxide methanol solution with a mass fraction of 20 g/L was added, a reflux condenser was connected to carry out saponification of fat and methyl esterification of fatty acids, and n-heptane was added, the oscillation was carried out for 2 min, and saturated sodium chloride aqueous solution was added. The upper solution was absorbed, and an appropriate amount of anhydrous sodium sulfate was added and shaken well, and the upper solution was absorbed into the injection bottle for machine determination. The machine conditions were as follows: the carrier gas was high-purity helium; the flow rate was 1.1 mL/min; the inlet and detector temperatures were 250 °C and 280 °C, respectively; the electron bombardment ion source (EI) was 70 eV; full scanning occurred; and the mass scanning range was 30–600 *m*/*z*.

### 2.5. Determination of Volatile Flavor Components

Headspace solid-phase microextraction and gas chromatography–mass spectrometry (GC-MS) combined technology were used to determine the volatile flavor components. The meat sample was cut into small pieces and crushed into minced meat, and 4 g of minced meat was weighed into a 20-milliliter headspace bottle, and 20% of NaCl particles and 5 μL of 2-methyl-3-heptanone solution (0.306 μg/mL) were added, shaken well, and heated in a water bath of 90 °C for 60 min, before being cooled to 60 °C. The SPME fiber head was aged at the GC-MS injection port, the adsorption head was inserted for 40 min of extraction, and the fiber was removed to desorb at the GC-MS injection port for 3 min via gas chromatography–mass analysis. The chromatographic conditions were as follows: the carrier gas was helium, the flow rate was 1.0 mL/min, the injection port temperature was 250 °C, the starting temperature of the column temperature was 40 °C, the temperature was maintained for 5 min, and the temperature was raised to 230 °C at 5 °C per minute for 8 min. The mass spectrometry conditions were as follows: the ion source temperature was 200 °C, the ionization mode was EI, the electron energy was 70 eV, and the scanning mass range was 35–500 amu.

### 2.6. RNA-seq Data Analysis

According to the manufacturer’s instructions, total RNA was extracted from ovine LTL using TRIzol^®^ reagent (Merck KGaA, Darmstadt, Germany). RNA samples were quantified on the basis of the A260/A280 absorbance ratio using a Nanodrop ND-2000 system (Thermo Scientific, Waltham, MA, USA), and the RIN of the RNA was calculated using an Agilent Bioanalyzer 4150 system (Agilent Technologies, Santa Clara, CA, USA). The PCR products were purified (AMPure XP system), and the quality of the libraries was evaluated via an Agilent Bioanalyzer 4150 system. Finally, paired-end reads of 150 bp were generated by sequencing the library preparations using an Illumina Novaseq 6000. Bioinformatic analysis was performed using the data generated via the Illumina (or BGI) platform.

### 2.7. Non-Targeted Metabolomics Analysis and Metabolite Identification

The muscle tissue was homogenized using the homogenizer, and 800 μL methanol/acetonitrile (1:1, *v*/*v*) was added to the homogenized solution for metabolite extraction. The supernatant was dried in a vacuum centrifuge, the samples were resuspended in 100 μL of acetonitrile/water (1:1, *v*/*v*) solvent and centrifuged at 14,000× *g* for 15 min at 4 °C, and the upper layer was injected for LC-MS analysis. Samples were analyzed via UHPLC (1290 Infinity LC, Agilent Technologies) combined with a quadruple time-of-flight (AB Sciex TripleTOF 6600). Mobile phase A was 25 mM of ammoniumacetate and 25 mM of ammoniumhydroxide in water, and B was acetonitrile. The gradient was 95% B for 0.5 min, before being linearly reduced to 65% in 6.5 min, further reduced to 40% in 1 min and held for 1 min, and then increased to 95% in 0.1 min, with a 3-min re-equilibration time. For MS-only acquisition, the instrument was configured to acquire data within the *m*/*z* range of 60 to 1000 Da, and the retention time for the product’s ion scan was configured to 0.05 ms/spectra. The data processing procedure was as follows: the raw MS data were converted into MzXML files using ProteoWizard MSConvert, before being imported into the freely available XCMS V9.1 software. Metabolites were identified via comparison with accurate *m*/*z* values (<10 ppm) and MS/MS spectra against an in-house database established using available authentic standards.

### 2.8. Real-Time Quantitative PCR

qRT-PCR was performed via the FOREGENE reverse transcription kit (RT OR-EasyTM II, FOREGENE, Cheng du, Sichuan, China) using a 20-microliter reaction system. The fluorescent quantitative PCR program and system were as follows: 95 °C for 2 min, 95 °C for 15 s, 58 °C for 30 s, 39 cycles of 95 °C for 15 s, and melt curve analysis (60~95 °C). GAPDH was used as a reference gene to normalize gene expression. The RNA OD value was detected via an ultra-micro nucleic acid protein analyzer (scandrop100), and the A260/A280 ratio was used. Fold-change was calculated for each candidate gene, and the sample was calculated using the 2^−ΔΔCT^ method. Quantitative data were presented as the mean ± standard error of the mean.

### 2.9. Statistical Analysis

The SPSS 22.0 software was used to perform the statistical analysis of the meat quality data. Means and standard errors of the mean (SEM) were obtained via one-way ANOVA and Duncan’s multiple comparisons. DEG and SCM were considered significantly correlated with a threshold of |r| > 0.5 and *p* < 0.05 and submitted to a biological conjoint annotation using the KEGG database. Pearson’s correlation coefficients were computed between SCM and DEGs through pair-wise comparison via the Hmisc package in R.

## 3. Results

### 3.1. Meat Quality of CK Sheep and TB Sheep

As indicated in Table 1, the obtained pH values after 45 min in the LTL of the CK and TB groups were within the normal range (6.2–6.5), and there was no difference in the meat color parameters (a* and b*). However, parameter L* of LTL was higher in the CK group than in the TB group (*p* < 0.05). Indicators such as moisture (MSTR), ash, crude protein (CP), and ether extract (EE) mainly reflect the conventional nutrients of meat. In our study, MSTR, ash, and CP were not different in these two breeds, but EE was higher in the LTL of TB sheep (*p* < 0.05).

Table 2 and Figure 1 depict the muscle fiber characteristics of the TB and CK sheep. The total number and density of muscle fiber in the CK group were higher than in the TB group (*p* < 0.05). The total areas and diameters of muscle fibers in these two breeds of sheep were similar.

### 3.2. The Composition of Amino Acids and Fatty Acids

The Amino acid (AAs) and Fatty acid (FAs) levels in these two different groups are shown in Table 3 and Table 4. The levels of EAAs, NEAAs, and TAAs in LTL were not different between the two groups. However, the glycine level was higher in CK sheep than in TB sheep (*p* < 0.05). No difference was found in SFA, SFA/UFA, PUFA/SFA, ω-3PUFA, ω-6PUFA, and ω-6/ω-3PUFA concentrations between the two groups according to the FA profile. Furthermore, TB sheep had higher PUFA and UFA levels than CK sheep (*p* < 0.05), but the content of (C22:6n3) was higher in CK sheep (*p* < 0.05).

### 3.3. Volatile Flavor Compounds of Meat

As shown in Table 5, 41 volatile flavor compounds were identified in the LTL of TB sheep and CK sheep, including 15 aldehydes, 7 alcohols, 10 ketones, 2 esters, and 7 other compounds. As shown in Table 5, the content of aldehydes (such as hexanal, heptanal, methylthiopropionaldehyde, etc.) in the LTL of CK sheep was higher (*p* < 0.05), but the alcohols, ketones, and esters were present in lower contents (*p* < 0.05). It is worth noting that the percentages of other volatile flavor compounds, such as 2-(pentenyl) furan, dimethyl disulfide, and dimethyl trisulfide, had no significant differences between the two groups.

### 3.4. Differentially Expressed Gene Identification and Functional Enrichment Analysis

To understand the differentially expressed genes in the LTL of these two breeds, we used comparative RNA-seq to analyze the transcriptomes of the LTL, and we obtained a total of 4.47 million high-quality clean reads via high-throughput sequencing and filtering of the raw reads for quality control. In the CK and TB groups, over 95% of the reads were mappable to the reference sheep genome. As shown in the volcanic diagram in Figure 2, we obtained 1009 DEGs for the TB vs. CK comparisons. Among these results, 853 genes were up-regulated and 156 genes were down-regulated in Tibetan sheep.

The functional enrichment analysis was performed to better understand the functions of the DEGs in the CK and TB groups (Figure 3). In this study, we concentrated on the top 30 enriched GO terms, including 10 biological process (BP) terms, 10 molecular function (MF) terms, and 10 cellular component (CC) terms. The enriched terms were related to cell growth, development, and metabolic processes related to meat quality, including cellular response to transforming growth factor beta stimulus, pyruvate metabolic process, regulation of small molecule metabolic process, defense response to symbiont (biological processes), steroid hormone receptor activity, phosphatidylinositol-3-phosphate binding, ornithine-oxo-acid transaminase activity and lipase binding (molecular functions), and protein–lipid complex and senescence-associated heterochromatin focus (cellular component). The KEGG enrichment results (Figure 4) indicated that the DEGs of these two groups were mostly enriched via pathways associated with the metabolic pathway, being especially enriched in amino acid and fatty acid-related metabolic pathways, such as arginine and proline metabolism, glutathione metabolism, and the biosynthesis of amino acids. The fatty acid-related metabolic pathways mainly included glycerophospholipid metabolism, sphingolipid signaling pathway, and cholesterol metabolism.

### 3.5. Metabolomic Profiling Based on UHPLC-Q-TOF MS

UHPLC-Q-TOF MS analysis was performed to study differential metabolites in the LTL of these two breeds of sheep. The quality control analysis demonstrated that the quality and quantity of the metabolomic assay data were sufficient for further metabolic expression analysis. OPLS-DA indicated that there were significant differences between TB sheep and CK sheep (Figure 5 and Figure 6).

We identified a total of 1325 DMs (718 in positive mode and 607 in negative mode), which we applied to perform multivariate analysis of metabolites. The 30 metabolites were classified into 13 superclasses, with the top six categories being lipids and lipid-like molecules (29.434%), organic acids and derivatives (22.943%), organic oxygen compounds (8.83%), organoheterocyclic compounds (8.453%), benzenoids (6.264%), and nucleosides, nucleotides, and analogs (4.679%). In comparison to the metabolites in the CK group, 95 differential metabolites significantly varied in the TB group, of which 37 were up-regulated and 58 were down-regulated. The biological mechanisms associated with the phenotypic changes were determined via KEGG enrichment analysis. The differential metabolites had enrichment in 158 biological pathways (*p* < 0.05). In the comparison of TB vs. CK, 38 differential metabolites were significantly enriched via eight pathways (*p* < 0.05), and most of them were related to lipid metabolism pathways, like glycerophospholipid metabolism, phospholipase D signaling pathway, aldosterone synthesis and secretion, arginine and proline metabolism, and ABC transporters (Figure 7 and Figure 8).

### 3.6. Integrative Analysis of the Transcriptome and Metabolome

In order to systematically analyze the differentially expressed genes and differential metabolites in the LTL of CK and TB sheep, we performed integrative analysis of the transcriptome and metabolome based on the results of the above transcriptome and metabolome studies. We performed KEGG pathway annotation and found that the main processes were related to amino acids and lipid metabolism pathways, including glutathione metabolism, taurine and hypotaurine metabolism, pentose phosphate pathway, glycerophospholipid metabolism, and the sphingolipid signaling pathway, which indicated that amino acid metabolism and lipid metabolism played essential roles in affecting the quality of the muscle in these two groups (Figure 9). The combined analysis of differentially expressed genes and differential metabolites showed that the sn-Glycerol 3-phosphoethanolamine was a key regulator, being responsible for lipid metabolism. As shown in the interactive network results, sn-Glycerol 3-phosphoethanolamine was an essential metabolite that was significantly associated with the expression of a variety of genes (*p* < 0.05) (Figure 10), including *ARAP3*, *TAPBP*, *MOGS*, *TIMP2*, *TSKS*, *ZNF205*, *ADCK5*, *ZSWIM8*, *SPTAN1*, and *CERS1*. It was also negatively correlated with *ATG5*, *RWDD2B*, and *TIGAR* expression. 5′-phosphoribosyl-5-amino-4-imidazolecarboxamide, which was related to amino acid metabolism, was positively correlated with the expression of *IGFBP3* and *PLCD1*. The differential metabolite alpha-L-Asp-L-Phe was positively correlated with the expression of *TBX2* and *LAMA5*.

### 3.7. Correlation Analysis of Meat Quality with DEGs and DMs

We conducted a correlation analysis between the significant DEGs enriched in the KEGG pathway and the key DMs (Figure 11). We found that the differentially expressed genes *GGT7*, *PFKP*, *GPAT2*, *PLPP2*, *AGPAT1*, *PNPLA2*, *MAPK12*, *NOS3*, *CERSA*, and *SMPD2* were associated with the key differential metabolites alpha-L-Asp-L-Phe, sn-Glycerol 3-phosphoethanolamine, D-mannose 6-phosphate, Isopentenyl pyrophosphate, Hydroxyacetone, 5′-phosphoribosyl-5-amino-4-imidazolecarboxamide, and Ile-Lys. Through the correlation analysis between DMs and differential meat quality, which was combined with KEGG significant metabolic pathway analysis, we speculated that the differential genes *GPAT2*, *PLPP2*, *AGPAT1*, and *PNPLA2* regulate differential metabolite Sn-Glycerol3-phosphoethanolamine and, through the glycerophospholipid metabolism pathway, further affect the volatile flavor substances (Aldehydes), the muscle fiber density of the LTL, and the meat color of sheep.

### 3.8. qRT-PCR Validation of Functional Gene Expression

To verify the authenticity and reliability of the transcriptome results, we randomly selected six DEGs and validated them via qRT-PCR. The results of validation experiments showed that the expression pattern of selected genes was consistent with the transcriptome results, which also indicated that the transcription result was true and reliable. We also found that the selected genes *ARMC2*, *ISG15*, *KLC3*, and *INAFM1* were significantly increased in CK sheep (Figure 12).

## 4. Discussion

There are differences in the meat quality of different breeds of animals [15]. The pH value is one of the important indicators of meat quality, as it can reflect the speed of muscle glycolysis and the level of lactic acid produced after slaughter [16]. In our study, the pH 45 min in the LTL of the CK and TB groups were within the normal range (6.2–6.5), and there was no significant differences in the pH values between these two samples when each one was measured 45 min after death. With the fermentation of muscle glycogen, a large amount of lactic acid accumulates, and the muscle PH value in CK and TB groups gradually decreases [17]. Previous studies have shown that meat color is an important indicator that consumers use to evaluate the freshness of meat products, which also reflects the antioxidant capacity of muscles, as is redness [18]. Value a* is positively correlated with the content of hemoglobin in muscles, yellowness value b* is negatively correlated with the freshness of meat, and brightness value L* is related to the whiteness of meat [19]. In this study, we found that there was no significant difference in the redness value a* and yellowness value b* between the CK group and the TB group (*p* > 0.05), while the L* value of the CK group was significantly higher than that of TB group (*p* < 0.05), which may be related to the difference in the antioxidant capacity of meat during muscle storage. Studies have shown that the color of meat is affected by myoglobin when the animal is slaughtered. Myoglobin exists in the form of deoxymyoglobin in the anaerobic state, and the content of myoglobin in the muscle of these two groups was different: the higher the content of oxidized muscle fibers, the brighter the meat of color, and the better the quality of meat [15]. Therefore, we infer that the LTL of Chaka sheep has more oxidative muscle fibers than that of Tibetan sheep, which may be caused by the higher brightness value L* in Chaka sheep. Meat quality, especially tenderness, flavor, and juiciness, are greatly affected by muscle fibers [20]. The total number and density were found to be significantly higher in the CK group than in the TB group in the present study, which indicated that the LTL of Chaka sheep had better tenderness and stronger water retention performance. Studies have identified that breed is the most crucial factor that influenced the characteristics of muscle fiber, including the types, sizes, and total numbers. The greater the density of the muscle fibers, the higher the economic value of the meat [21]. This result may be one of the reasons why the meat of the Chaka sheep is such a consumer favorite.

The nutritional indicators of meat are essentially the moisture content; the proportions of ash, protein and fat; and the composition of FAs and AAs. [22]. In our study, the contents of MSTR, Ash, and CP were not significantly different between the Chaka sheep group and the Tibetan sheep group, while the amount of EE, UFA, and PUFA were significantly lower in Chaka sheep than in Tibetan sheep (*p* < 0.05). Fat is an important nutrient in the animal’s body, and it is positively correlated with the flavor of meat. The types and amounts of fatty acids in different kinds of sheep are different, and the characteristics of fat mainly depend on fatty acids [23]. Different types of fatty acids will produce different flavor substances after oxidation and decomposition, which will affect the flavor of the meat. Unsaturated fatty acid has a high melting point and high density, which helps to improve the antioxidant capacity of meat [24]. Our study found that the content of UFA in the TB group was higher than in the CK group, which further suggests that the LTL of Tibetan sheep may have a higher antioxidant capacity than that of Chaka sheep. Vitor et al. [25] have shown that when the content of polyunsaturated fatty acids in lamb increases, the color and flavor of the meat will decrease, which further affects consumers’ choices. PUFAs are easy to oxidize, and when the oxygen content is too high, it is easy to cause oxidative rancidity of the meat, which produces a peculiar smell, thereby reducing the quality of the meat [26]. The results of this study found that the content of PUFA in the LTL of Tibetan sheep was higher than in that of Chaka sheep. The more PUFAs in Tibetan sheep may be easily oxidized, and when the oxygen content is too high, it may cause oxidative rancidity of the meat, which produce a peculiar smell, thereby reducing the quality of Tibetan sheep meat. The ratio of n-6/n-3 is a significant parameter used to evaluate the nutritional value of meat [27]. To better maintain cardiovascular health, the n-6/n-3 ratio in the human diet should not exceed 4 [28]. In our study, the n-6/n-3 ratio in the LTL of Tibetan sheep was well above 4, exceeding the Food and Agriculture Organization of the United Nations’ (FAO) recommendation [29], but the ratio was closer to 4 in Chaka sheep. This result provides further evidence that the consumption of Tibetan lamb is not good for cardiovascular health.

Protein is an essential nutrient in the human body, as it can provide energy for the body, participate in the composition of the body, and catalyze a variety of biochemical reactions in the body [30]. Amino acids are substrates involved in protein synthesis, and their content and composition can reflect the nutritional value of muscles. A previous study has shown that lamb is a good source of protein because its skeletal muscle contains a variety of essential amino acids needed by the human body, which can improve the body’s immunity and reduce morbidity and mortality [31]. In this study, there was no significant difference (*p* > 0.05) in total amino acids and essential amino acids in the LTL of Chaka sheep and Tibetan sheep, which indicated that the two breeds of sheep have good nutritional value. Protein and amino acids are not only essential nutrients, but they also have an effect on the taste of meat products [32]. Previous studies have shown that free amino acids acting as precursors have an effect on the flavor of meat, which is important in evaluating the nutritional quality of meat [33]. Our study found that the content of glycine in the LTL of Chaka sheep was significantly higher than that of Tibetan sheep (*p* < 0.05). Some AAs are the main determinants of the taste of mutton. For example, glycine often gives mutton a sweet taste, which is favored by consumers [33]. It is believed that the improvement in the levels of these amino acids is important for the quality of meat [34]. Therefore, we speculated that the higher glycine content in Chaka sheep may give Chaka sheep meat a better flavor. From this point of view, the flavored amino acid in the LTL of Chaka sheep was higher than in that of Tibetan sheep, and the flavor of Chaka sheep meat was better.

Previous studies have shown that the amino acid composition of meat is closely correlated to the flavor and nutritional value of meat, while the fatty acid content is a determinant of the nutritional value and antioxidative stability of muscle and especially significant for the quality of meat and the acceptance of meat products [35]. Our results were consistent with this finding. In this study, we found that the flavor of the longest dorsal muscle of the Chaka sheep was better than that of the Tibetan sheep, which had higher contents of fat and fatty acids (UFA and PUFA) than the Chaka sheep. The content of glycine was significantly higher in Chaka sheep than in Tibetan sheep, and the unique sweetness of glycine imparts a better flavor to the meat, whereas too high a content of polyunsaturated fatty acids tends to oxidize and produce a peculiar smell, which further affects the flavor of the meat [33]. The higher unsaturated fatty acid content in Tibetan sheep favors their antioxidant properties. In conclusion, amino acids and fatty acids, as two important parameters involved in meat quality, are closely related to the nutritional quality of meat on the one hand, and on the other hand, they affect the flavor and texture of meat. Some essential amino acids and polyunsaturated fatty acids cannot be synthesized by the human body and must be obtained from food [28,29]. There is no significant difference in the content of essential fatty acids between Chaka sheep and Tibetan sheep, but the content of polyunsaturated fatty acids in Chaka sheep is significantly lower than that of Tibetan sheep, which means that the fatty acid content of Tibetan sheep has a higher nutritional value.

As the flavor precursors in meat produce a variety of volatile flavor compounds through complex chemical reactions that occur during processing, they give the meat its unique odor [36]. Different kinds of compounds contribute differently to the flavor of the meat, and the volatile flavor is a vital parameter for evaluating meat quality [37]. The volatile flavor substances detected in sheep typically include aldehydes, alcohols, and ketones, a finding that is consistent with the results of previous studies. In this study, we detected a total of five volatile flavor substances in two breeds of sheep, including 15 aldehydes, 7 alcohols, 10 ketones, 2 esters, and 7 other compounds. Alcohol compounds are derived from the degradation of lipids, which have a high odor threshold and a generally clear fragrance akin to the smell of fruits and vegetables. The ester compounds are formed via the esterification of acid and alcohol, and most of them have an aromatic taste. Aldehyde compounds are produced via fatty acid oxidation and amino acid degradation, with a low odor threshold and greater impact on the flavor of meat, but excessive oxidation of fatty acids will deteriorate and harm the flavor of meat [38]. We found that the contents of alcohols, ketones, and esters in the Chaka sheep were significantly lower than those in the Tibetan sheep (*p* < 0.05), while the contents of aldehydes were significantly higher than those in the Tibetan sheep (*p* < 0.05), which indicated that the ability of the Chaka sheep to deposit flavor substances was weak. Aldehydes are a significant group of fragrance and flavor substances, and certain concentrations of aldehydes will help to produce good aromas. For example, a suitable concentration of hexanal gives meat apple and leafy aromas, while heptanal gives meat nutty and fruity aromas [11]. Our study found that the hexanal and heptanal contents in the LTL of Chaka sheep were significantly higher than those of Tibetan sheep (*p* < 0.05), and this observation may be the reason that Chaka sheep meat has a better flavor and is preferred by consumers. Nonanal has a pronounced fatty and sour taste. In addition, alcohols have less effect on lamb flavor than aldehydes. However, in high levels, they can yield vanilla, woody, and fatty flavors [39]. We found that the overall content of alcohols in the LTL of Tibetan sheep was significantly higher than that in Chaka sheep (*p* < 0.05), which may give Tibetan sheep a more pronounced fatty taste, and our previous results showed that the crude fat content in the LTL of the Tibetan sheep is higher, which may also lead to the taste of the meat of the Tibetan sheep being greasier than that of the Chaka sheep. Lipids and fatty acids are the most important sources of volatile flavor compounds in meat [23]. Meanwhile, alcohols and ketones are the major oxidation and degradation components in lipids and FAs [13]. Therefore, this observation may also be the reason for the higher fat and fatty acid content in Tibetan sheep.

We combined transcriptomics and metabolomics analysis to identify differential genes, differential metabolites, and differential metabolic pathways that affect meat quality. Our findings showed that the main processes were related to amino acids and lipid metabolism pathways, such as glutathione metabolism, taurine and hypotaurine metabolism, glycerophospholipid metabolism, and sphingolipid signaling pathway, which indicated that amino acid metabolism and lipid metabolism played extensive and crucial roles in the quality of the two breeds of sheep muscle. Acetyl phosphate and glutamic acid were significant differential metabolites enriched in the taurine and hypotaurine metabolic pathways, and *GGT7* (gamma-glutamyltransferase 7) is the main differentially expressed gene enriched via this pathway, which is upregulated in Chaka sheep. In our study, the significant differential metabolites enriched in the glutathione metabolic pathway mainly include L-pyroglutamic acid, glutamic acid, and glutathione, and the differential metabolites glutamate and glutathione had up-regulated expressions in CK sheep. Studies have shown that glutamate is an important taste-presenting amino acid, and glutathione is a tripeptide that is high in muscle and essential for muscle metabolism and homeostasis [40]. The up-regulation of L-glutamate and glycine can further promote the sweet and umami tastes to contribute to the improvement in the flavor of meat [41]. Meanwhile, glutathione has an essential role in radical scavenging, detoxification, and signal transduction [38]. In addition, taurine and hypotaurine, being two sulfur-containing amino acids, play vital roles in metabolic pathways [40]. The reaction of sulfur-containing amino acids with sugars produces many sulfur-containing compounds that affect the flavor of the meat [42]. Therefore, we speculated that it was these differential amino acids and peptides that have a certain regulatory effect on the flavor of sheep LTL by participating in muscle metabolism and giving Chaka sheep meat a better flavor.

In addition to amino acid metabolism, lipid metabolism is also a significant differential metabolic pathway in our study. Previous research has shown that in addition to nutrients, lipids are important precursors of food flavors, especially their derivatives [43]. The influence of phospholipids on the production of flavor compounds in meat was found to be much greater than that of triglycerides [44]. In our study, glycerophospholipid metabolism and the sphingolipid signaling pathway are two significant differential metabolic pathways, among which sn-glycerol 3-phosphoethanolamine is the key differential metabolite involved in upregulated expression in Chaka sheep. *GPAT2* (glycerol-3-phosphate acyltransferase 2), *PLPP2* (phospholipid phosphatase2), AGPAT1 (1-acylglycerol-3-phosphate-O-acyltransferase1), *PNPLA2* (phospholipase domain containing 2), and *GPAT4* (glycerol-3-phosphate acyltransferase 4) were the main differentially expressed genes enriched via this pathway, which are all upregulated in Chaka sheep. Previous studies have shown that the *PNPLA2* gene was found to encode an enzyme that is involved in catalyzing the first step involved in the catabolism of triglycerides in adipose tissue [45,46]. Through subsequent correlation analysis of significant differential metabolites and differentially expressed genes, we found that these genes had significant positive correlations with the differential metabolites glycerol 3-phosphate and sn-Glycerol 3-phosphoethanolamine. In addition, we performed a correlation analysis regarding significant differential metabolites and differential meat quality, and the results showed that sn-glycerol 3-phosphoethanolamine had a significant positive correlation with meat color, muscle fiber density, and volatile flavor substances in lamb [47]. Phospholipids are the main source of volatile flavors and phospholipids, including glycerophospholipids and sphingomyelin, and sn-glycerol 3-phosphoethanolamine is closely related to terpenoids [48]. In addition, aldehydes and alcohols are also regulated by metabolites derived from lipids. A study conducted by Zhang [49] shows that regulation of lipid metabolism, including glycerophospholipid metabolism, etheric lipid metabolism, and glycerol lipid metabolism, could significantly enhance the taste of Tibetan sheep muscle. Therefore, we speculated that *GPAT2*, *PLPP2*, *AGPAT1*, *PNPLA2*, and *GPAT4* in the LTL of Chaka sheep show better meat color, muscle fiber density, and flavor than those of Tibetan sheep by regulating the expression of sn-Glycerol 3-phosphoethanolamine through the glycerophospholipid metabolism pathway.

## 5. Conclusions

In summary, this study demonstrated that variety has a certain effect on meat quality. Based on meat quality analysis, the nutritional characteristics of meat, such as the EE, UFA, and PUFA of Tibetan sheep, were better, while the texture and flavor of Chaka sheep were better. Moreover, a large number of DEGs and DMs were identified on the basis of the integrative analysis of transcriptomics and metabolomics in the muscles of Chaka sheep and Tibetan sheep. The main differential metabolic pathways involved in the two breeds were related to amino acid metabolism (particularly glutathione metabolism, taurine, and hypotaurine metabolism) and lipid metabolism-related pathways (such as glycerophospholipid metabolism and sphingolipid signaling pathway), and most importantly, the intermediate metabolite sn-Glycerol 3-phosphoethanolamine played a key role during phospholipid metabolism, which may be regulated by *GPAT2*, *PLPP2*, *AGPAT1*, *PNPLA2*, and *GPAT4*, as well as positively correlated with the meat quality (flavor, color, and texture). Overall, these results not only provide strong scientific support for future research into meat quality, but also offer guidance and reference to studies of the selection of key genes and metabolites associated with muscle growth and meat quality.

## Figures and Tables

**Figure 1 animals-13-02494-f001:**
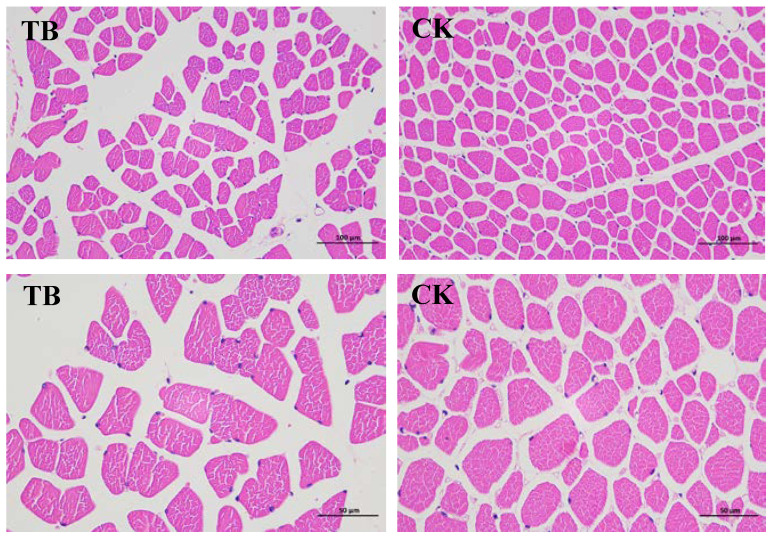
The muscle fiber characteristics in LTL. TB, Tibetan sheep group; CK, Chaka sheep group.

**Figure 2 animals-13-02494-f002:**
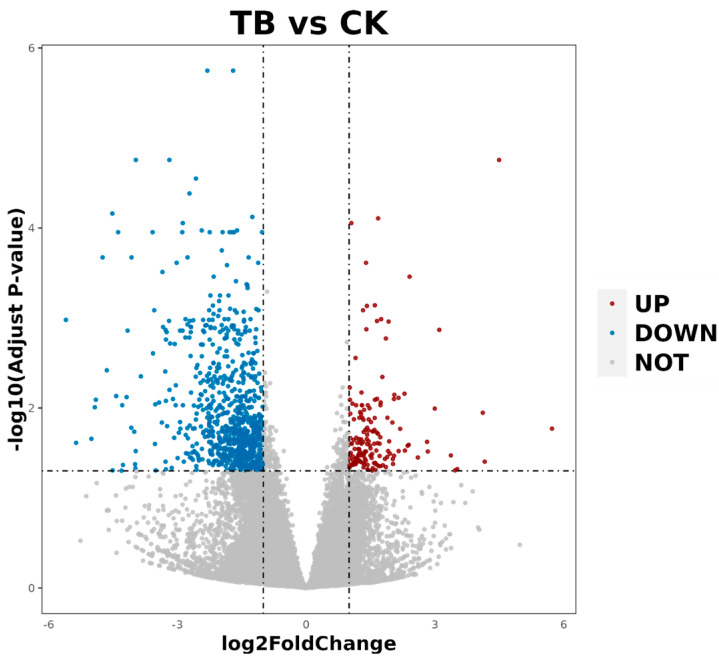
Differential gene expression distribution volcano map.

**Figure 3 animals-13-02494-f003:**
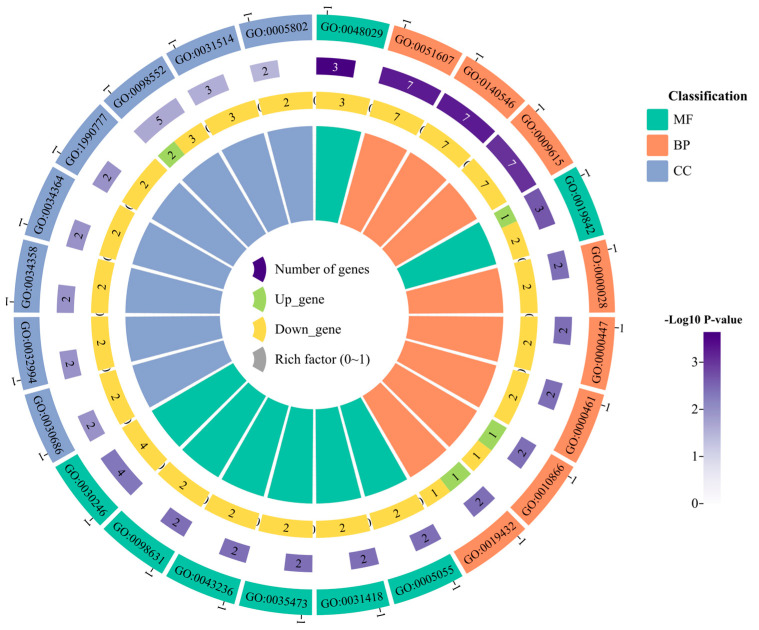
Differential gene GO enrichment map.

**Figure 4 animals-13-02494-f004:**
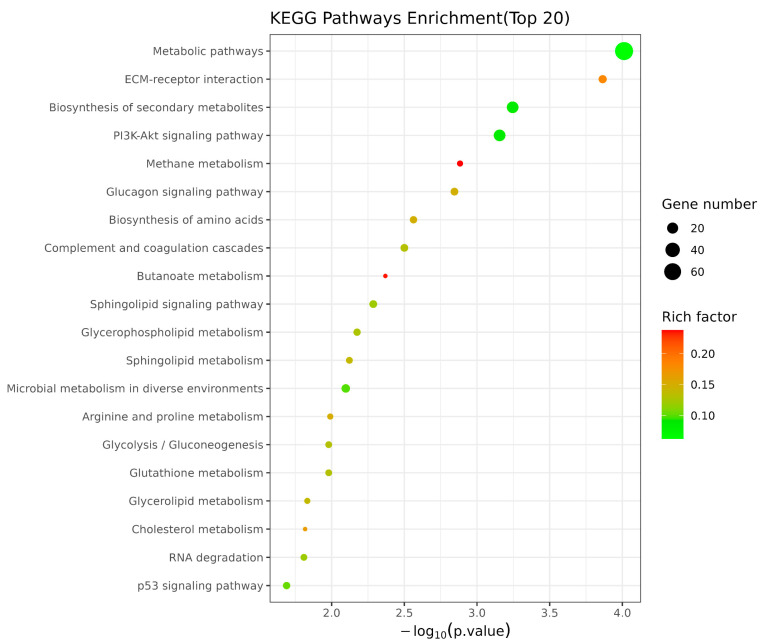
KEGG enrichment bubble plot of differentially expressed genes.

**Figure 5 animals-13-02494-f005:**
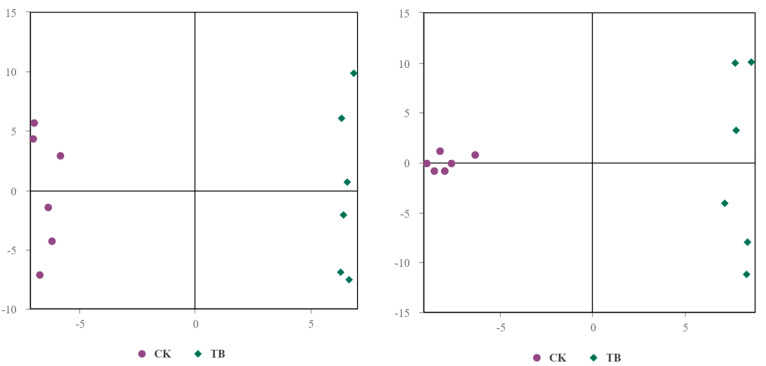
The OPLS-DA score map in positive and negative ion modes.

**Figure 6 animals-13-02494-f006:**
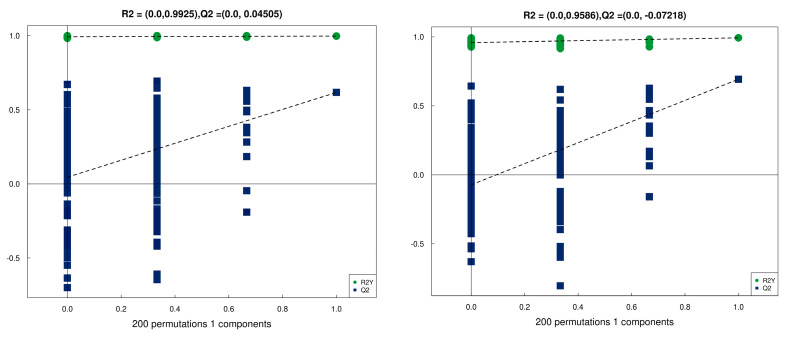
The OPLS-DA displacement test in positive and negative ion modes.

**Figure 7 animals-13-02494-f007:**
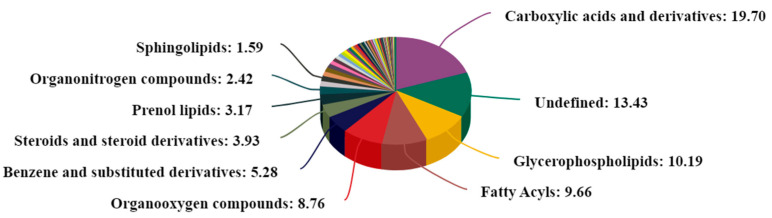
Quantitative proportion of identified metabolites in each chemical class.

**Figure 8 animals-13-02494-f008:**
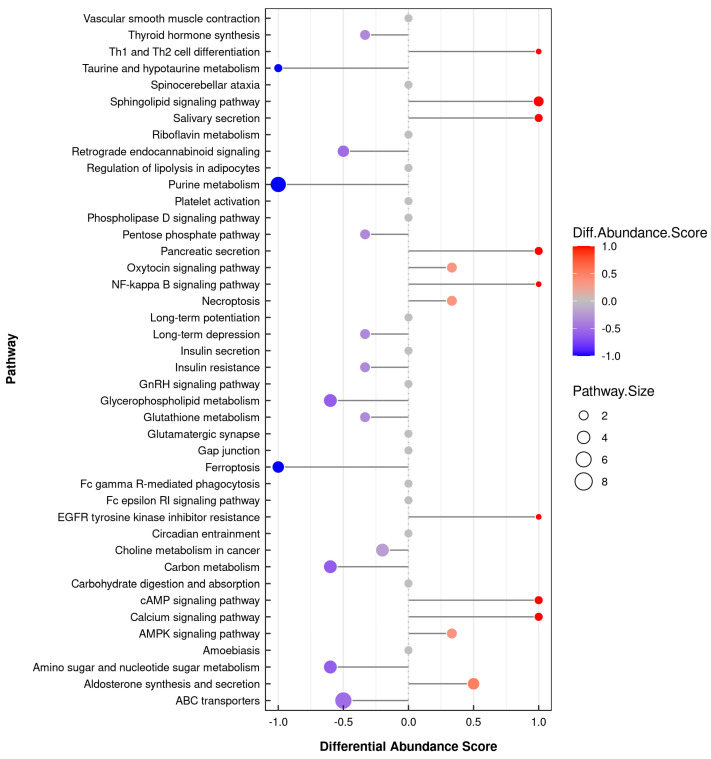
Differential abundance score plots of all differential metabolic pathways.

**Figure 9 animals-13-02494-f009:**
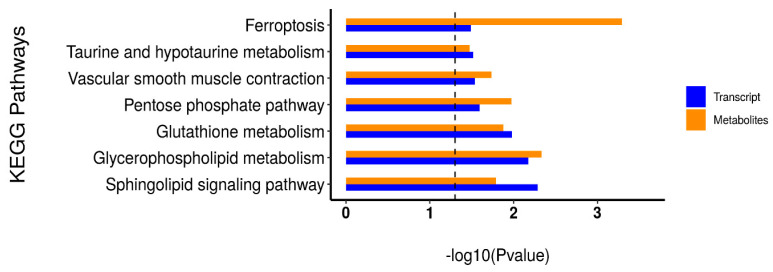
KEGG enrichment histogram of differential genes and metabolites.

**Figure 10 animals-13-02494-f010:**
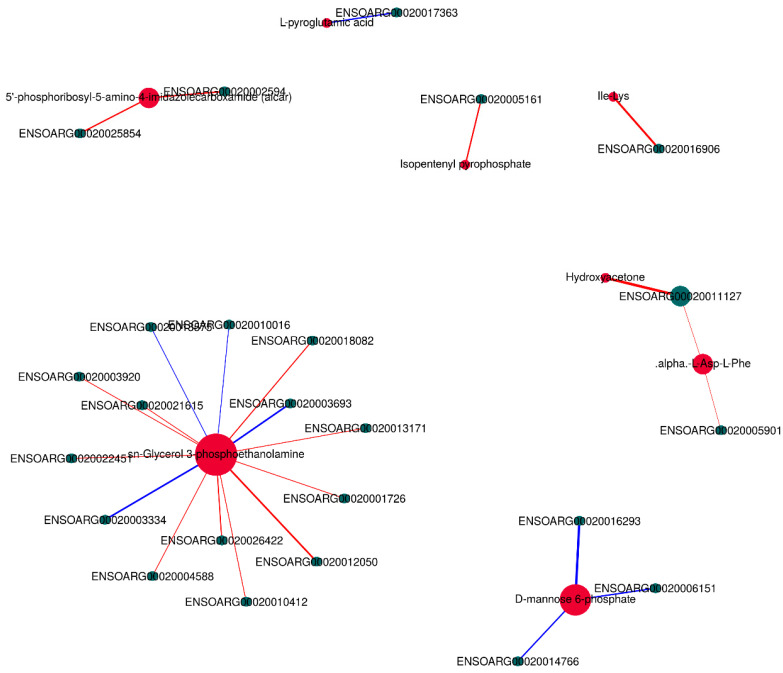
Correlation analysis network diagram of significant differential genes and significant differential metabolites.

**Figure 11 animals-13-02494-f011:**
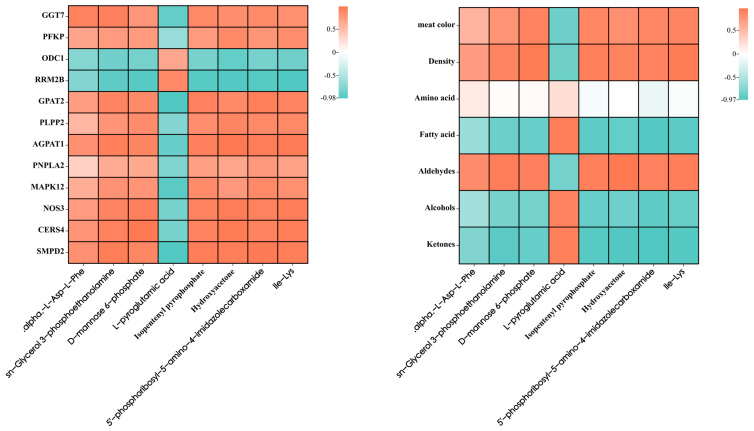
Correlation analysis between significant DMs with DEGs and meat quality.

**Figure 12 animals-13-02494-f012:**
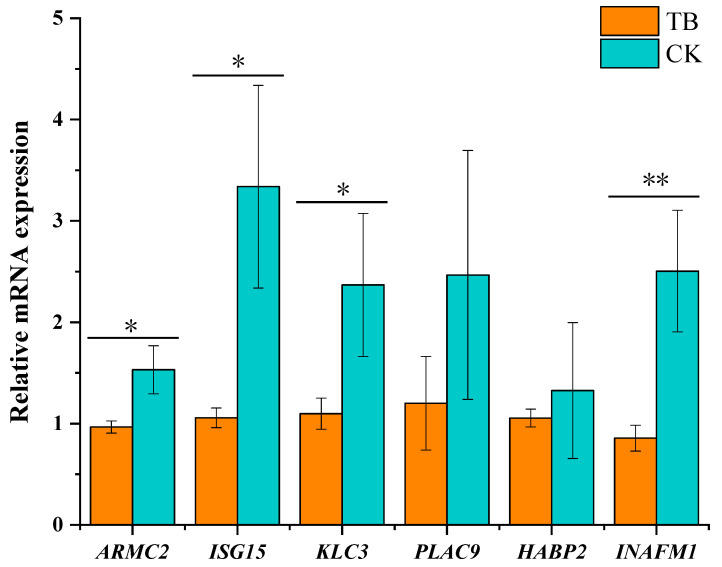
Validation of the expression patterns of the six candidate genes via qRT-PCR. Notes: * indicates significant difference at *p* < 0.05 and ** indicates highly significant difference at *p* < 0.01.

**Table 1 animals-13-02494-t001:** Conventional meat quality and nutritional parameters of the LTL of Chaka sheep and Tibetan sheep.

Items	CK ^1^	TB ^2^	SEM ^3^	*p*-Value
pH	6.24 ± 0.055	6.50 ± 0.164	0.074	0.058
L*	31.67 ± 1.155	28.33 ± 1.528	0.894	0.039
a*	29.67 ± 2.082	24.67 ± 2.517	1.400	0.057
b*	13.00 ± 2.000	13.00 ± 2.000	0.730	1.000
MSTR (%)	71.50 ± 0.700	73.07 ± 1.464	0.546	0.170
Ash (%)	1.99 ± 0.137	2.17 ± 0.033	0.053	0.101
EE (%)	10.56 ± 0.367	11.95 ± 0.574	0.331	0.024
CP (%)	22.63 ± 0.378	23.50 ± 0.721	0.286	0.139

Notes: ^1^ CK, Chaka sheep group; ^2^ TB, Tibetan sheep group; ^3^ SEM, standard error of the mean. Ash, CP, and EE contents are expressed on a DM basis. L*, Luminance, a*, Redness, b*, Yellowness.

**Table 2 animals-13-02494-t002:** The muscle fiber characteristics of Tibetan and Chaka sheep.

	CK ^1^	TB ^2^	SEM ^3^	*p*-Value
total number	61.67 ± 2.082	44.67 ± 7.234	4.269	0.017
total area, (mm^2^)	0.04 ± 0.001	0.04 ± 0.006	0.002	0.293
density, (n/mm^2^)	1426.54 ± 43.804	1158.77 ± 109.356	67.157	0.017
diameter, (mm)	0.03 ± 0.003	0.03 ± 0.001	0.0008	0.388

Notes: ^1^ CK, Chaka sheep group; ^2^ TB, Tibetan sheep group; ^3^ SEM, standard error of the mean.

**Table 3 animals-13-02494-t003:** Amino acid content in LTL of Chaka sheep and Tibetan sheep.

Items (%)	CK ^1^	TB ^2^	SEM ^3^	*p*-Value
Asp	5.72 ± 0.265	5.66 ± 0.196	0.086	0.756
Thr	2.95 ± 0.173	3.02 ± 0.095	0.053	0.590
Ser	2.32 ± 0.170	2.42 ± 0.040	0.049	0.407
Glu	11.53 ± 0.644	12.12 ± 0.270	0.223	0.220
Gly	3.05 ± 0.040	2.88 ± 0.064	0.043	0.018
Ala	3.79 ± 0.156	3.79 ± 0.086	0.046	0.976
Cys	0.34 ± 0.074	0.36 ± 0.038	0.022	0.697
Val	3.32 ± 0.159	3.36 ± 0.153	0.057	0.807
Met	1.14 ± 0.150	0.92 ± 0.122	0.070	0.124
Ile	3.32 ± 0.140	3.39 ± 0.137	0.053	0.570
Leu	5.38 ± 0.250	5.44 ± 0.150	0.076	0.753
Tyr	2.13 ± 0.270	2.10 ± 0.079	0.073	0.878
Phe	3.56 ± 0.255	3.49 ± 0.158	0.079	0.694
His	2.76 ± 0.157	2.64 ± 0.136	0.060	0.362
Lys	5.73 ± 0.253	5.75 ± 0.203	0.084	0.933
Arg	4.37 ± 0.187	4.33 ± 0.090	0.054	0.736
Pro	2.55 ± 0.064	2.58 ± 0.026	0.019	0.543
TAA	63.97 ± 2.873	64.23 ± 1.535	0.843	0.894
EAA	28.17 ± 1.344	28.01 ± 0.862	0.414	0.871
NEAA	35.80 ± 1.540	36.56 ± 0.401	0.445	0.455
IMP	1.67 ± 0.060	1.79 ± 0.160	0.052	0.292

Notes: ^1^ CK, Chaka sheep group; ^2^ TB, Tibetan sheep group; ^3^ SEM, standard error of the mean.

**Table 4 animals-13-02494-t004:** Fatty acid content in LTL of Chaka sheep and Tibetan sheep.

Items (%)	CK ^1^	TB ^2^	SEM ^3^	*p*-Value
SFA	48.83 ± 2.272	49.29 ± 2.381	0.856	0.820
(C4:0)	0.02 ± 0.018	0.03 ± 0.006	0.005	0.611
(C6:0)	0.01 ± 0.002	0.02 ± 0.009	0.003	0.233
(C8:0)	0.13 ± 0.007	0.02 ± 0.008	0.003	0.622
(C10:0)	0.15 ± 0.006	0.19 ± 0.039	0.013	0.176
(C11:0)	0.001 ± 0.002	0.003 ± 0.0002	0.0007	0.238
(C12:0)	0.06 ± 0.0081	0.09 ± 0.0193	0.007	0.139
(C13:0)	0.01 ± 0.0001	0.01 ± 0.008	0.002	0.260
(C14:0)	2.10 ± 0.112	2.61 ± 0.604	0.200	0.202
(C15:0)	0.28 ± 0.006	0.34 ± 0.053	0.019	0.147
(C16:0)	24.49 ± 0.856	27.00 ± 1.611	0.732	0.076
(C17:0)	1.03 ± 0.043	1.06 ± 0.015	0.013	0.447
(C18:0)	20.21 ± 2.359	17.52 ± 0.290	0.859	0.122
(C20:0)	0.06 ± 0.027	0.09 ± 0.011	0.011	0.122
(C21:0)	0.25 ± 0.016	0.18 ± 0.042	0.018	0.071
(C22:0)	0.11 ± 0.021	0.07 ± 0.014	0.010	0.076
(C23:0)	0.05 ± 0.006	0.06 ± 0.029	0.008	0.560
(C24:0)	0.02 ± 0.006	0.02 ± 0.004	0.003	0.195
MUFA	41.72 ± 3.542	45.49 ± 3.888	1.446	0.051
(C14:1)	0.11 ± 0.003	0.16 ± 0.045	0.016	0.145
(C15:1)	0.16 ± 0.007	0.16 ± 0.036	0.009	0.859
(C16:1)	1.29 ± 0.207	1.45 ± 0.105	0.069	0.307
(C17:1)	0.48 ± 0.057	0.54 ± 0.027	0.021	0.183
(C18:1n9t)	ND	ND	-	-
(C18:1n9c)	39.41 ± 3.330	42.90 ± 4.054	1.463	0.461
(C20:1)	0.11 ± 0.019	0.13 ± 0.032	0.011	0.513
(C22:1n9)	0.02 ± 0.013	0.01 ± 0.005	0.004	0.846
(C24:1)	0.14 ± 0.019	0.10 ± 0.032	0.013	0.150
PUFA	3.97 ± 0.568	5.71 ± 0.630	0.228	0.047
(C18:2n6t)	ND	0.94 ± 0.514	-	-
(C18:2n6c)	3.30 ± 0.439	2.53 ± 0.029	0.205	0.040
(C18:3n6)	0.54 ± 0.146	0.58 ± 0.181	0.061	0.770
(C18:3n3)	0.15 ± 0.018	0.12 ± 0.041	0.013	0.383
(C20:2)	ND	0.01 ± 0.012	-	-
(C20:3n3)	0.26 ± 0.020	0.22 ± 0.028	0.012	0.131
(C20:4n6)	0.08 ± 0.012	0.06 ± 0.017	0.008	0.079
(C20:3n6)	0.87 ± 0.037	0.63 ± 0.144	0.065	0.052
(C22:2)	0.02 ± 0.0023	0.01 ± 0.0062	0.002	0.208
(C20:5n3)	0.65 ± 0.111	0.56 ± 0.076	0.041	0.285
(C22:6n3)	0.11 ± 0.035	0.04 ± 0.014	0.018	0.039
UFA	47.69 ± 2.985	51.20 ± 3.798	1.366	0.042
SFA/UFA	0.97 ± 0.103	1.03 ± 0.118	0.043	0.535
PUFA/SFA	0.12 ± 0.006	0.17 ± 0.018	0.005	0.629
ω-3PUFA	1.17 ± 0.154	0.94 ± 0.145	0.074	0.140
ω-6PUFA	4.78 ± 0.555	4.74 ± 0.501	0.193	0.927
ω-6/ω-3PUFA	4.15 ± 0.738	5.06 ± 0.462	0.303	0.144

Notes: ^1^ CK, Chaka sheep group; ^2^ TB, Tibetan sheep group; ^3^ SEM, standard error of the mean. ND indicates not detected.

**Table 5 animals-13-02494-t005:** Contents of volatile flavor compounds in LTL of CK sheep and TB sheep.

Items (%)	CK ^1^	TB ^2^	SEM ^3^	*p*-Value
Aldehydes	62.94 ± 5.997	40.58 ± 2.158	5.264	0.04
Pentanal	0.24 ± 0.051	0.34 ± 0.122	0.041	0.261
2-Thiophene formaldehyde	0.21 ± 0.059	0.34 ± 0.082	0.384	0.095
Hexanal	4.28 ± 0.144	2.58 ± 0.354	0.393	0.002
Heptanal	4.67 ± 0.879	1.65 ± 0.195	0.714	0.004
Octanal	3.65 ± 0.445	1.66 ± 0.693	0.493	0.014
Nonanal	13.78 ± 0.677	3.26 ± 0.335	2.361	<0.001
Methylthiopropionaldehyde	ND	0.89 ± 0.305	-	-
Decanal	1.39 ± 0.300	1.25 ± 0.099	0.087	0.471
Benzaldehyde	21.24 ± 3.623	18.49 ± 1.625	1.196	0.296
(E)-2-Decenal	2.17 ± 0.089	1.57 ± 0.833	0.255	0.283
2-Undecenal	3.72 ± 0.498	1.54 ± 0.180	0.506	0.002
2-Phenyl-2-butylenaldehyde	0.79 ± 0.055	0.50 ± 0.070	0.688	0.005
(E,E)-2,4-heptadienal	0.73 ± 0.305	0.19 ± 0.023	1.431	0.039
Pentadecanal	1.26 ± 0.350	1.02 ± 0.241	0.123	0.372
Hexadecanal	9.35 ± 0.291	5.54 ± 0.390	0.861	<0.001
Alcohols	6.60 ± 0.801	8.30 ± 0.131	0.433	0.022
1-Pentanol	0.78 ± 0.078	1.07 ± 0.262	0.095	0.143
1-Hexanol	0.44 ± 0.099	0.57 ± 0.101	0.047	0.185
2-Furanmethanol	0.37 ± 0.097	0.46 ± 0.145	0.049	0.422
1-Octen-3-ol	1.55 ± 0.387	2.51 ± 0.351	0.255	0.033
1-Heptanol	0.93 ± 0.119	1.32 ± 0.315	0.123	0.115
1-Octanol	1.86 ± 0.211	1.89 ± 0.297	0.094	0.893
(E)-2-Octene-1-ol	0.67 ± 0.72	0.44 ± 0.154	0.068	0.079
Ketones	1.72 ± 0.180	3.34 ± 0.264	0.372	0.001
2-Pentadecanone	0.58 ± 0.085	0.65 ± 0.145	0.462	0.511
3-Hexanone	0.13 ± 0.045	0.75 ± 0.232	0.152	0.011
2,3-Octanedione	0.78 ± 0.082	0.37 ± 0.056	0.095	0.002
Hydroxyacetone	ND	1.21 ± 0.078	-	-
6-Methyl-5-hepten-2-one	0.23 ± 0.061	0.36 ± 0.042	0.035	0.035
Hydrocarbons	3.78 ± 0.700	5.80 ± 2.471	0.802	0.246
Heptylbenzene	0.99 ± 0.125	1.92 ± 0.405	0.234	0.020
Xylene	0.76 ± 0.371	0.89 ± 0.396	0.143	0.699
Octylbenzene	1.17 ± 0.154	1.68 ± 0.055	0.122	0.006
Nonylbenzene	0.86 ± 0.146	1.64 ± 0.331	0.198	0.020
Esters	0.47 ± 0.129	1.98 ± 0.572	0.370	0.011
2-Ethylethyl acetate	ND	0.24 ± 0.127	-	-
Methyl diethyldithiocarbamate	0.47 ± 0.129	1.73 ± 0.446	0.308	0.009
Others	9.94 ± 1.023	17.29 ± 4.920	2.095	0.064
2-Pentyl-furan	1.50 ± 0.280	4.76 ± 1.006	0.777	0.006
2-Acetyl thiazole	0.15 ± 0.025	0.58 ± 0.125	0.100	0.005
2- (pentenyl) furan	1.40 ± 0.627	1.21 ± 0.362	0.192	0.673
2-Acetyl pyrrole	1.49 ± 0.148	3.26 ± 0.180	0.400	<0.001
Acetoin	3.60 ± 0.246	1.69 ± 0.242	0.436	0.001
Dimethyl disulfide	0.18 ± 0.030	0.20 ± 0.173	0.010	0.374
Dimethyl trisulfide	0.27 ± 0.021	0.25 ± 0.087	0.024	0.719

Notes: ^1^ CK, Chaka sheep group; ^2^ TB, Tibetan sheep group; ^3^ SEM, standard error of the mean. ND indicates not detected

## Data Availability

Not applicable.

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
