# Peer review of "Evaluation of the *Longissimus Thoracis et Lumborum* Muscle Quality of Chaka and Tibetan Sheep and the Analysis of Possible Mechanisms Regulating Meat Quality"

_animals, 2023, doi:10.3390/ani13152494_

Round 1
Reviewer 1 Report
Dear Authors
The manuscript submitted for review is interesting and worthy of publication with the proposed revisions. The title is fully consistent with the content of the manuscript. In the "Simple Summary" section-please do not use abbreviations unless they are explained earlier in the content-see line 11. In the "Abstract" (this applies to the entire text of the manuscript as well) the authors do not use the required spaces between words in many places, please correct this. In the "Keywords"-section, there is longissimus dorsi, and there should be m. longissimus dorsi- preferably in italics, as with all Latin names. In the "Introduction" section...please correct the sentence "...the concept of a reasonable and healthy diet is more and more deeply rooted in the hearts of the people...". please do not use colloquialisms in scientific texts. Please explain why the authors chose Chaka sheep and Tibetan sheep as the object of their scientific experiment? Please characterize for the readers Chaka sheep and Tibetan sheep, give their origin, when they were created? are they unique to the region from which they originated?, maybe they are endemic? please include for the readers photos of sample representatives of these breeds. Section " Materials and Methods", correct in line 92: longissimus dorsi muscle. Please post , for non-specialists, the topographical location of m.longissimus dorsi in the photo. The results are presented with due care, the tables and figures are well described. The discussion is interesting to the reader and the results are promising.
Regards
Minor editing of English language required.
Author Response
Dear Reviewer
We feel great thanks for your professional review work on our article. As you are concerned, there are several problems that need to be addressed. According to your nice suggestions, we have made extensive corrections to our previous draft, the detailed corrections are listed below.
Q1: In the "Simple Summary" section-please do not use abbreviations unless they are explained earlier in the content-see line 11.
We sincerely thanks for your careful reading. As suggested by you, we have corrected the abbreviations”TB and CK” into “Tibetan and Chaka”.
Q2: In the "Abstract" (this applies to the entire text of the manuscript as well) the authors do not use the required spaces between words in many places, please correct this.
We feel sorry for our carelessness. In the resubmitted manuscript, we've made corrections. Thanks for your correction.
Q3: In the "Keywords"-section, there is longissimus dorsi, and there should be m. longissimus dorsi- preferably in italics, as with all Latin names.
Thanks for your suggestion. As suggested by you, we have corrected the ”longissimus dorsi” into “m. longissimus dorsi” in the “Keywords” section.
Q4: In the "Introduction" section...please correct the sentence "...the concept of a reasonable and healthy diet is more and more deeply rooted in the hearts of the people...". please do not use colloquialisms in scientific texts.
Thanks for your suggestion. We have tried our best to correct the sentence in the revised manuscript.
Q5: Please explain why the authors chose Chaka sheep and Tibetan sheep as the object of their scientific experiment? Please characterize for the readers Chaka sheep and Tibetan sheep, give their origin, when they were created? are they unique to the region from which they originated?, maybe they are endemic? please include for the readers photos of sample representatives of these breeds.
We think this is an excellent suggestion. We have supplemented the manuscript (in line 56-70) by reviewing the literature to the best of our ability.Secondly, in response to your question about providing readers with photos of different breeds of sheep, we did our best to find previous information, but realized that we were in a hasty sampling process and didn't have pictures at the time that would be suitable for use in the article. Your opinion is a very good one, but we really can't make up the photos later, so we ask for your understanding!
Q6: Section " Materials and Methods", correct in line 92: longissimus dorsi muscle. Please post , for non-specialists, the topographical location of m.longissimus dorsi in the photo.
Thanks for your careful checks. We are sorry for our carelessness. Based on your comments, we have made the corrections in the manuscript, Since we don't have photos of the carcasses of the animals, we can't mark them, but we wrote in the article about the specific part of the sample that was collected, so we ask for your understanding!You've made a very good suggestion, and we'll definitely take it up next time and make sure to photograph it!
We tried our best to improve the manuscript and made some changes marked in yellow in revised paper which will not influence the content and framework of the paper. We appreciate for your warm work earnestly, and hope the correction will meet with approval. Once again, thank you very much for your comments and suggestions.
The author of this manuscript

Reviewer 2 Report
The work is very interesting and original since it aims to evaluate meat quality of two sheep breeds and to identify potential bio-markers through transcriptomics and metabolics correlations analysis. This approach was used hoping find possible genes, differential metabolites and significant differential metabolic pathways explaining meat quality differences between breeds.
Overall, the manuscript is well presented and written. However, some remarks should be taken into consideration:
1. The Absract
1.1. some data must be added as quality parameters (pH, protein, EE, ash contents ....)
1.2. Some given results in the abstract are not true and different from results : (line 22 : protein and amino acids are higher in TB sheep than in CK one. Results reported in table 1 indicated that CP was not diferent and the only difference in AA profil was the glycine on that was higher in CK breed and not in TB as you wrote.
2. The Introduction
2.1. The firts sentence (lines 35-37) must be improved.
2.2. The multi-omics linkage technique, used in this work, was not well developed in the introduction since only two references (12 & 13) were cited. The advantages and the limits of this approach should be highlighted.
2.3. Line68 : put a point at the end of the sentence; before the word Studies have shown...
3. Material and Methods
3.1. Data of age, weight and sexmust be added since these factors affect directly meat quality
4. Results
4.1. Lines 204-205 : "In our study, there aws no difference in MSTR and CP content but the fat and protein content of TB was higher": CP was indicated twice with different results (NS and higher???)
4.2. I don't understand why the sum of moisture, ash, CP and EE was greater than 100 %, unless the contents of ash, CP and EE are expressed as DM basis; if yes please indicate it the the table 1.
4.3. Line 212 : Remove the end of the sentence "difference is not significant" since you wrote just before "are similar"
4.4. In table 4, C18:2n6t and C20:2 were ND for CK sheep and you calculated the significance between the two breeds ??? How ?
4.5. Lines 221-223 : "there was no difference in the SFA, PUFA........", however C22:6n3 was significantly higher in CK than in TB breed
4.6. In figure 7 : the undefined represented 13.43% of the metabolites, which was very high since it represented the second most "component". Is thishigh value did not affect the results and the the reliability of ypur analysis ?
4.7. Line 377 : "6 DEGs were randmoly chosen", is this approach didn't also affect the results ? some alternatives methods of choice ?
5. Discussion
5.1. Line 375 : remove the word "loved" using other word more appropriate
5.2. Line 380 : "the amount of EE and TFA were significantly lower in CK than TB breed" ; this TFA was not given any where in results
5.3. Lines 419-422 : "In CK sheep, the content of glycine wassignificantly higher than TB sheep and aspartic acid, methionine....were higher but the difference was not significant"; results showed no differences , we can't use differently these resultas since some differences are very low : tyrosine (2.13 vs 2.10), phenylalanine (3.56 vs 3.49) and arginine (4.37 vs 4.33). This discussion must be reformulated
5.4. In the discussion, the effects of AA and FA on the meat quality are well discussed separately but not combined; this part can be easily improved
5.5. Lines 464-466 : the main processes were related to amino acids and lipid metabolism, which is very interestingand. But the only difference on AA profil in your study concerned glycine. Is it enough to explain the importance of AA or the role of lipid (FA ?) was more important (preponderant) in these mechanisms ?
5.6. Your results are not well compared to other ones using the same methods
English can be improved
Author Response
Dear Reviewer
We feel great thanks for your professional review work on our article. As you are concerned, there are several problems that need to be addressed. According to your nice suggestions, we have made extensive corrections to our previous draft, the detailed corrections are listed below.
The Absract
Q1: some data must be added as quality parameters (pH, protein, EE, ash contents ....)
We sincerely thanks for your careful reading. As suggested by you, we have added differential meat quality parameters to the abstract.
Q2: Some given results in the abstract are not true and different from results : (line 22 : protein and amino acids are higher in TB sheep than in CK one. Results reported in table 1 indicated that CP was not diferent and the only difference in AA profil was the glycine on that was higher in CK breed and not in TB as you wrote.
We sincerely thanks for your valuable feedback, and this section has been reviewed and revised to include the parameters with significant variation (crude fat, unsaturated and polyunsaturated fatty acids, and parameters related to muscle fiber properties) and to excuse the inadequate analysis of the results due to the less significant results of the meat quality parameters.
The Introduction
Q3: The firts sentence (lines 35-37) must be improved.
Thanks for your suggestion. We have tried our best to correct the sentence in the revised manuscript.
Q4: The multi-omics linkage technique, used in this work, was not well developed in the introduction since only two references (12 & 13) were cited. The advantages and the limits of this approach should be highlighted.
We think this is an excellent suggestion. We have supplemented the manuscript (in line 72-76) obout the advantages and the limits of multi-omics linkage technique.
Q5: Line68 : put a point at the end of the sentence; before the word Studies have shown...
Thanks for your careful checks. We are sorry for our carelessness. Based on your comments, we have made the corrections in the manuscript.
Material and Methods
Q6: Data of age, weight and sex must be added since these factors affect directly meat quality
We fully agree with your proposal, and have supplemented the data of age, weight and sex in the “Material and Methods” section.
Results
Q7: Lines 204-205 : "In our study, there was no difference in MSTR and CP content but the fat and protein content of TB was higher": CP was indicated twice with different results (NS and higher???)
We were really sorry for our careless mistakes. Thank you for your reminder. In the manuscript, we have revised this section as shown in lines 207-208 of the manuscript.
Q8: I don't understand why the sum of moisture, ash, CP and EE was greater than 100 %, unless the contents of ash, CP and EE are expressed as DM basis; if yes please indicate it the table 1.
Thank you for your reminder, the contents of ash, CP and EE were expressed as DM basis,we have indicated in the table 1.
Q9: Line 212 : Remove the end of the sentence "difference is not significant" since you wrote just before "are similar"
We feel sorry for our carelessness. In our resubmitted manuscript, the sentence was revised. Thanks for your correction.
Q10: In table 4, C18:2n6t and C20:2 were ND for CK sheep and you calculated the significance between the two breeds ??? How ?
We feel sorry for our carelessness,by checking and verifying the experimental data, we found that there were a few data entry errors during data entry due to our negligence, which we have corrected in the form.Thanks for your careful checks.
Q11: Lines 221-223 : "there was no difference in the SFA, PUFA........", however C22:6n3 was significantly higher in CK than in TB breed
First of all, we apologize for our carelessness, there was an error in entering some of the data in this part of the data entry process, resulting in a mismatch between the resultant analysis and the experimental data, which we have corrected, as shown in lines 226-228 of the manuscript. We sincerely thank you for your careful checking and correction.
Q12: In figure 7 : the undefined represented 13.43% of the metabolites, which was very high since it represented the second most "component". Is this high value did not affect the results and the the reliability of your analysis ?
Thank you for asking an interesting question. The undefined represented 13.43% of the metabolites were just some compounds of the identified metabolites that are not clearly categorized, it is not certain which category they belong to, and this figure is just to illustrate the situation of the metabolites that we have co-identified and categorized. In the later analyse, we looked for differential metabolic pathways in the muscles of the two breeds of sheep with the help of KEGG pathway enrichment analysis and analyzed the differential metabolites enriched in the differential metabolic pathways, specifically analyzing which differential metabolites they were, so that the lack of clarity on the broad class to which they belonged should not affect the reliability of the results of the subsequent analyses.
Q13: Line 377 : "6 DEGs were randmoly chosen", is this approach didn't also affect the results ? some alternatives methods of choice ?
Thank you for your question. To verify the authenticity and reliability of the transcriptome results, we randomly selected six DEGs and validated them via qRT-PCR. This section is designed to validate the amount and expression of genes in a transcriptomics study, so the validated genes should be randomly selected in the validation exercise, rather than purposefully selecting particular genes. In the subsequent analysis, we analyzed the differentially expressed genes related to meat quality rather than these six validated genes selected randomly, so we think it had no effect on our experimental results.
Discussion
Q14: Line 375 : remove the word "loved" using other word more appropriate
Thanks for your suggestion, we have replaced the word "loved" by “ favorite”.
Q15: Line 380 : "the amount of EE and TFA were significantly lower in CK than TB breed" ; this TFA was not given any where in results
TFA in the article means Total Fatty Acid Content, not Trans Fatty Acid, in the previous results we did the sum of Total Fatty Acid, and then later expressed the fatty acid content in percentage, but neglected to modify the corresponding text part, thank you for your detailed review, we have modified the content of the part as you suggested.
Q16: Lines 419-422 : "In CK sheep, the content of glycine wassignificantly higher than TB sheep and aspartic acid, methionine....were higher but the difference was not significant"; results showed no differences , we can't use differently these resultas since some differences are very low : tyrosine (2.13 vs 2.10), phenylalanine (3.56 vs 3.49) and arginine (4.37 vs 4.33). This discussion must be reformulated
This issue we fully agree and accept your suggestion to analyze and discuss only the significantly different glycine in the manuscript and remove the discussion of the non-significantly different parameters, thank you very much for your valuable suggestion.
Q17: In the discussion, the effects of AA and FA on the meat quality are well discussed separately but not combined; this part can be easily improved
Thanks to your suggestion, we have added the effect of amino acids and fatty acids on meat quality to the manuscript.
Q18: Lines 464-466 : the main processes were related to amino acids and lipid metabolism, which is very interestingand. But the only difference on AA profil in your study concerned glycine. Is it enough to explain the importance of AA or the role of lipid (FA ?) was more important (preponderant) in these mechanisms ?
You raised an interesting question.In this study, we firstly determined the macro-indicators related to meat quality, and our results revealed that the content of glycine, unsaturated fatty acids and polyunsaturated fatty acids differed significantly between the two breeds of sheep back longest muscle. Secondly, we investigated the differential expression at the molecular level in the muscles of the two breeds by means of histological linkage techniques. Through combined transcriptomic and metabolomic analyses, we found that the differentially expressed genes and differentially metabolized products were mainly enriched in the amino acid metabolism pathway and the lipid metabolism pathway, among which one metabolite in the glycerophospholipid metabolism pathway sn-glycerol 3-phosphoethanolamine was found to be the most important metabolite in the metabolism pathway. phosphoethanolamine was correlated with 14 genes, and through the later correlation analysis, we found that the differential metabolite was related to meat color, flavor, and texture, so we hypothesized that the differential metabolite had a certain effect on meat quality.
In addition, the difference in macroscopic data such as glycine and fatty acids is only an indication of the phenotypic manifestation of the meat of the two breeds, and it is not used to justify the influence of amino acid metabolism and lipid metabolism on meat quality. The metabolic mechanisms related to meat quality were analyzed by histological techniques, and we cannot determine whether amino acid metabolism or lipid metabolism has a greater influence on meat quality, but we can only say that an intermediate product of glycerophospholipid metabolism, sn-glycerol 3-phosphoethanolamine, may play an important role in the regulation of the color and flavor of meat. flavor. This question is worthy of further study in subsequent research.
Q19: Your results are not well compared to other ones using the same methods
We agree and gladly accept your evaluation. However, we do not think that a significant difference is the best result. Although our results, as you said, did not systematically and comprehensively reveal a relatively large scientific problem, we also made some new discoveries at a small level, at least clarified the specific differential meat indexes in the muscles of the two breeds of sheep, and analyzed and revealed the causes of the differences in some way, which is of some reference value and guidance for the subsequent studies. In addition, we are also aware of similar studies by others, and their experimental design may focus more with short-term supplemental feeding, such as high-energy feeds, whereas the only variable in our trial was the breed, so it also suggests that supplemental feeding has a more significant effect on the quality of lamb meat.
We tried our best to improve the manuscript and made some changes marked in yellow in revised paper which will not influence the content and framework of the paper. We appreciate for your warm work earnestly, and hope the correction will meet with your approval. Once again, thank you very much for your comments and suggestions!
The author of this manuscript
